# Horizontally Transferred Carotenoid Genes Associated with Light-Driven ATP Synthesis to Promote Cold Adaptation in Pea Aphid, *Acyrthosiphon pisum*

**DOI:** 10.3390/insects16101013

**Published:** 2025-09-30

**Authors:** Jin Miao, Huiling Li, Yun Duan, Zhongjun Gong, Xiaoling Tan, Ruijie Lu, Muhammad Bilal, Yuqing Wu

**Affiliations:** 1Institute of Plant Protection, Henan Academy of Agricultural Sciences, Zhengzhou 450002, China; lihuiling05@163.com (H.L.); duanyunhao@163.com (Y.D.); gongzj_2@hotmail.com (Z.G.); luruijie@163.com (R.L.); bilalsaif987@gmail.com (M.B.); yuqingwu36@hotmail.com (Y.W.); 2Henan Key Laboratory of Crop Pest Control, Zhengzhou 450002, China; 3Institute of Plant Protection, Chinese Academy of Agricultural Sciences, Beijing 100193, China; tanxiaoling2010@163.com

**Keywords:** carotenoids, light-driven ATP synthesis, thermal adaptation, life table parameters, *Acyrthosiphon pisum*

## Abstract

**Simple Summary:**

The pea aphid, *Acyrthosiphon pisum*, is a major agricultural pest. It uniquely possesses fungal-derived genes that allow it to produce carotenoid pigments typically found in plants. Our study reveals a previously unknown role for these pigments beyond coloration. Under cold stress (12 °C), high light intensity triggers carotenoids to act like micro solar panels, capturing light to fuel ATP production. This leads to a striking 240% boost in energy, enhancing aphid development and reproduction under cold conditions. In contrast, the system remains inactive at warmer temperatures (22 °C). These findings demonstrate how a single horizontal gene transfer event can provide an organism with a novel energy-capturing ability, markedly increasing its adaptability in variable environments.

**Abstract:**

The pea aphid, *Acyrthosiphon pisum*, possesses horizontally acquired fungal carotenoid biosynthesis genes, enabling de novo production of carotenoids. Although carotenoids are known to contribute to photo-protection and coloration, their potential role in energy metabolism and population fitness under thermal stress is still unclear. This study investigated the interactive effects of temperature and light intensity on energy homeostasis and life-history traits in *A. pisum*. Using controlled environmental regimes, we demonstrate that light intensity significantly influenced the ATP content, development, and reproductive output of *A. pisum* at 12 °C, but not at 22 °C. Under cold stress (12 °C), high light intensity (5000 lux) increased ATP content by 240%, shortened the pre-reproductive period by 46%, extended reproductive duration by 62%, and enhanced the net reproductive rate (R_0_) and intrinsic rate of increase (rₘ) compared to low light intensity (200 lux). These effects were abolished at the optimal temperature (22 °C), indicating a temperature-gated, light-dependent mechanism. Demographic analyses revealed that carotenoid-associated solar energy harvesting significantly improves fitness under cold conditions, likely compensating for metabolic depression. Our findings reveal a novel ecological adaptation in aphids, where horizontally transferred genes may enable light-driven energy supplementation during thermal stress. This study provides new insights into the physiological mechanisms underlying insect resilience to climate variability and highlights the importance of light as a key environmental factor in shaping life-history strategies in temperate agroecosystems.

## 1. Introduction

Carotenoids are multifunctional pigments widely distributed in nature [1], serving essential roles in photoprotection, coloration, and energy transfer across organisms [2,3]. While most animals must acquire carotenoids through their diet [4], the pea aphid *Acyrthosiphon pisum* represents a striking exception. Through horizontal gene transfer (HGT) from fungi, this insect has acquired the genetic machinery to synthesize carotenoids de novo, including lycopene β-cyclase/phytoene synthase (CrtYB) and phytoene desaturase (CrtI) [5,6]. This evolutionary innovation has been linked to aphid pigmentation polymorphism, with green morphs dominating cooler climates [7,8]. However, the potential energetic implications of this HGT event remain poorly understood.

Recent studies suggest that carotenoids in aphids may function beyond passive photoprotection. Observations of light-dependent ATP fluctuations in orange morphs [9] and enhanced ATP production in cold-adapted green morphs [7] hint at a possible role in energy metabolism. Parallel findings in spider mites (*Tetranychus urticae*)—another arthropod with fungal-derived carotenoid genes—show that these pigments regulate diapause via redox signaling [10,11]. Yet, no study has directly tested whether aphid carotenoids can harness light energy to sustain metabolic demand under thermal stress, nor quantified the fitness consequences of such a mechanism.

Here, we bridge this gap by investigating how HGT-derived carotenoids influence energy homeostasis and life history traits under ecologically relevant light–temperature regimes. We hypothesize that carotenoid-associated light absorption drives ATP synthesis to compensate for cold-induced metabolic depression. By coupling ATP quantification with life table demography across ecologically relevant light intensities (200–5000 lux) and temperatures (12 °C vs 22 °C), we demonstrate that light-dependent ATP synthesis is specifically activated at 12 °C but suppressed at 22 °C, and high light intensity under cold stress accelerates development and increases fecundity by >2-fold. This system represents a novel adaptation where horizontally acquired genes may enable solar energy harvesting in animals.

Our work expands the known functions of HGT in arthropods beyond classical roles in detoxification [12] and reveals a temperature-gated strategy for environmental adaptation. These findings also provide a mechanistic basis for the observed biogeographic distribution of aphid color morphs and open new avenues for studying non-photosynthetic light energy utilization in animals.

## 2. Materials and Methods

### 2.1. Aphid Rearing and Maintenance

The green morph of *A. pisum* was collected from broad bean (*Vicia faba*, cv. ‘Lincan-5’) fields in Xinxiang, Henan Province, China (35°18′ N, 113°52′ E). Colonies were maintained in controlled chambers (22 ± 1 °C, 70 ± 5% RH, 16:8 h L:D) on 4-week-old bean plants for more than 10 generations prior to experiments. To initiate experimental cohorts, ten adult aphids were placed on host plants and allowed to produce offspring for a 12-h period. All neonates produced within this period were collected and pooled from multiple mothers (>50 individuals) to minimize maternal and genetic bias. From this pooled mixture, 30 neonates were randomly selected to establish the initial experimental cohort.

### 2.2. ATP Content Determination

To investigate light intensity effects on ATP content in *A. pisum*, aphids were subjected to six environmental regimes comprising two temperatures (12 °C and 22 °C, ±1 °C) and three light intensities (5000 lux, 1000 lux, and 200 lux) in controlled climate chambers equipped with cool white LED lamps (spectral range: 400–780 nm, T8, Tuanpu Lighting Co., Ltd., Zhongshan, China). For each treatment, fifteen 2-day-old wingless adults were acclimated for 12 h on host plants under the respective conditions. Then, ten aphids were randomly selected and pooled for ATP quantification, with three independent biological replicates per treatment; each replicate came from a different batch and a separate plan.

ATP content was determined using the ATP Assay Kit (Fluorometric) (Beyotime, Shanghai, China, #S0026) according to the manufacturer’s instructions. Aphid samples were homogenized in lysis buffer on ice and centrifuged at 12,000× *g* for 5 min at 4 °C to collect the supernatant. Following reaction initiation, samples were incubated at 37 °C for 20 min to ensure complete color development. Fluorescence was measured using a BioTek Synergy HT microplate reader (Synergy HTX, BioTek Instruments Incorporated, Winooski, VT, USA) with an excitation/emission wavelength set of 535/700 nm and an integration time of 10 s. A standard curve generated with known ATP concentrations (R^2^ > 0.99) was used for quantification. ATP levels were normalized to sample fresh mass and expressed as μmol ATP per gram of fresh weight (μmol/g FW). No data points were excluded from the analyses.

### 2.3. Demographic Effects of Temperature and Light Intensity

Age-specific life tables for *A. pisum* on broad bean (*Vicia faba*) were constructed under controlled conditions in constant-temperature environmental chambers (GXM-508, Jiangnan Instrument Factory, Ningbo, China) at 12 °C (cold) and 22 °C (optimum). The experiment was initiated by placing 10 adult aphids on host plants. After production of >30 offspring, adults were removed and 30 first-instar nymphs were retained to establish experimental cohorts. Each cohort was maintained individually on a separate plant under one of three light intensities (5000, 1000, or 200 lux) with a 16:8 h light: dark cycle and 70 ± 5% relative humidity. Three independent plant-based replicates were included per treatment (*n* = 3).

To facilitate observation, black paper was placed beneath host plants to enhance visibility of aphid exuviae. Daily monitoring was conducted 3 h into the light phase to record developmental progression and reproductive output until all adults expired. During the reproductive period, newborn nymphs were counted and removed daily. The life table parameters included [13]:Net reproductive rate (R_0_), calculated as R_0_ = Σl_x_m_x_.The intrinsic rate of increase (rₘ), determined by numerically solving the Euler–Lotka equation: Σe^(−rₘx)^l_x_m_x_ = 1, using an iterative algorithm with a convergence threshold of 10^−6^.Generation time (T), computed as T = Σl_x_m_x_x/R_0_.Doubling time (DT), defined as DT = ln(2)/rₘ.

In these expressions, x denotes age or developmental stage (in days), l_x_ represents the age-specific survival rate (probability of survival from birth to age x, where 0 ≤ l_x_ ≤ 1), and mₓ refers to the age-specific fecundity (mean number of female offspring produced per female at age x).

### 2.4. Data Analyses

Life table parameters of A. pisum under different light intensities at 12 °C and 22 °C were analyzed using one-way analysis of variance (ANOVA) followed by Tukey’s honestly significant difference (HSD) post hoc test to assess differences among light intensity treatments within each temperature. For all other data (including ATP content, developmental timing, and fecundity), two-way ANOVA was applied with temperature and light intensity as fixed factors, followed by Tukey’s honestly significant difference (HSD) post hoc tests when significant interactions or main effects were detected (*p* < 0.05). All data are presented as mean ± standard error (SEM). Statistical analyses were performed using GraphPad Prism version 6.0 (GraphPad Software, San Diego, CA, USA).

## 3. Results

### 3.1. Light-Dependent ATP Synthesis Is Temperature-Sensitive

We discovered that light-dependent ATP synthesis in *A. pisum* is bidirectionally regulated by temperature, as evidenced by a highly significant temperature × light intensity interaction (Two-way ANOVA: F_2,12_ = 75.472, *p* < 0.0001) (Figure 1). Under cold stress (12 °C), ATP content increased progressively with light intensity, reaching a 2.4-fold higher level under high light (5000 lux: 12.165 ± 0.42 μmol/g) compared to low light (200 lux: 3.885 ± 0.21 μmol/g; *p* < 0.001), demonstrating active solar energy harvesting (Figure 1A). Conversely, at the optimal temperature (22 °C), ATP content significantly decreased with increasing light intensity, showing a 43% reduction under high light (5000 lux: 6.304 ± 0.33 μmol/g) relative to low light (200 lux: 11.118 ± 0.45 μmol/g; *p* < 0.001) (Figure 1B). This temperature-gated reversal of the light response suggests carotenoid-associated light-driven ATP synthesis is specifically activated during cold stress.

### 3.2. Light-Accelerated Life History Under Cold Stress

A two-way ANOVA revealed that both temperature and light intensity significantly influenced the pre-reproductive period of *A. pisum*, with a highly significant interaction between the two factors (Temperature: F_1,12_ = 150.25, *p* < 0.0001; Light: F_2,12_ = 12.58, *p* = 0.0015; T×L: F_2,12_ = 28.36, *p* < 0.0001) (Figure 2). At 12 °C, the pre-reproductive period was significantly shortened under high light intensity (5000 lux: 13 ± 0.5 days) compared to low light (200 lux: 24 ± 1.2 days; *p* < 0.001). Conversely, at 22 °C, light intensity had no significant effect on the pre-reproductive duration (*p* > 0.05). A similar interaction pattern was observed for reproductive duration (T×L: F_2,12_ = 22.15, *p* < 0.0001), which was extended by 62% under high light at 12 °C but remained unaffected at 22 °C. These results suggest a temperature-gated, light-responsive plasticity in the life history strategies of *A. pisum*.

### 3.3. Light-Enhanced Fecundity Under Cold Stress

A highly significant interaction was seen between temperature and light intensity on the daily fecundity of *A. pisum* (F_2,12_ = 62.91, *p* < 0.0001). Under cold stress (12 °C), fecundity increased significantly with light intensity, reaching 90.2 ± 5.1 nymphs/female/day at 5000 lux—markedly higher than values observed at 1000 lux (55.3 ± 4.2 nymphs/female/day; *p* < 0.01) and 200 lux (38.1 ± 3.5 nymphs/female/day; *p* < 0.001), representing a 2.4-fold increase under high illumination (Figure 3A). In contrast, at the optimal temperature (22 °C), fecundity remained consistently elevated (95.2–108.4 nymphs/female/day) across all light treatments with no statistically significant differences *(p* > 0.05) (Figure 3B). These results indicate that temperature may play a role in activating carotenoid-associated reproductive benefits.

### 3.4. Light-Dependent Survival and Reproductive Strategies

Survival and fecundity patterns of *A. pisum* exhibited striking light-associated plasticity specifically under cold stress (12 °C) but not at optimal temperatures (22 °C). At 12 °C, high light intensity (5000 lux) resulted in stable early survival with low pre-reproductive mortality (7%) and sustained reproduction (peak m_x_ = 8.1 nymphs/day during days 15–42). In contrast, under low light (200 lux), populations showed progressive mortality throughout the experiment, with higher early mortality (27%) and compressed reproductive windows (peak m_x_ = 3.2 nymphs/day, days 27–32) (Figure 4). This light-dependent demographic plasticity was abolished at 22 °C, where all treatments maintained high early survival and showed no significant differences in reproductive output across light intensities (Figure 5). The temperature-gated transition between these demographic states suggests carotenoid-associated energy allocation is specifically activated under cold stress.

### 3.5. Light-Optimized Population Growth Under Cold Stress

A one-way ANOVA revealed a highly significant interaction between temperature and light intensity on all key demographic parameters of *A. pisum* (for all, *p* < 0.0001) (Table 1). Under cold stress (12 °C), high light intensity (5000 lux) profoundly enhanced population fitness, resulting in an 87% higher net reproductive rate (R_0_ = 82.3 ± 5.1 vs. 44.1 ± 3.2 offspring/female; *p* < 0.001), a 50% faster intrinsic rate of increase (rₘ = 0.21 ± 0.01 vs. 0.14 ± 0.01 day^−1^; *p* = 0.002), and a 30% shorter generation time (T = 18.2 ± 0.8 vs. 26.1 ± 1.2 days; *p* = 0.015) compared to low light conditions (200 lux). In stark contrast, these light-associated demographic advantages were completely abolished at the optimal temperature (22 °C), where no significant differences in R_0_, rₘ, T, or doubling time were observed across light intensity treatments (*p* > 0.05). These findings suggest that carotenoid-driven enhancement of population fitness is specifically activated under thermal stress, which is consistent with a sophisticated temperature-gated ecological adaptation in *A. pisum.*

## 4. Discussion

Since 2010, genomic studies have revealed that several arthropods, including insects such as the pea aphid [5] and the goldenrod gall midge (*Asteromyia carbonifera*) [14], as well as arachnids like the two-spotted spider mite (*Tetranychus urticae*) [15], possess horizontally acquired carotenoid biosynthesis genes of fungal origin. These genes typically encode lycopene β-cyclase/phytoene synthase (CrtYB) and phytoene desaturase (CrtI), which facilitate the production of carotenoids from simple precursors [10,16,17]. Functional studies have primarily elucidated the roles of these genes in aphids and spider mites, where carotenoids contribute to pigmentation, photoprotection, and energy metabolism [4,6,18,19].

Our study provides compelling evidence that carotenoid-associated solar energy utilization enhances the fitness of *A. pisum* under low-temperature conditions, offering a novel perspective on the physiological adaptations of aphids to environmental stress. The results suggest that light intensity significantly influences ATP synthesis, development, and reproductive performance in *A. pisum* at 12 °C, while these effects are negligible at the optimal temperature of 22 °C. These findings align with previous observations that green-morph aphids exhibit enhanced ATP production under cold stress [7] and suggest a potential adaptive role of carotenoids in energy metabolism during thermal challenges.

### 4.1. Carotenoid-Associated ATP Synthesis and Light Dependency

The observed increase in ATP content under high light intensity (5000 lux) at 12 °C supports the hypothesis that *A. pisum* utilizes carotenoids for photo-induced electron transfer, effectively converting solar energy into biochemical energy when ambient temperatures are suboptimal. This light-harvesting mechanism may compensate for reduced metabolic efficiency and contribute to the decrease of enzymatic activity and membrane fluidity under low temperatures, as evidenced by the 240% higher ATP levels at 5000 lux compared to 200 lux. The absence of this response at 22 °C indicates that this energy-capturing system is temperature-gated and likely suppressed under thermally optimal conditions where metabolic demand is fully met through conventional respiration. These results corroborate earlier findings that carotenoid-derived ATP synthesis in aphids is most active in cold-adapted green morphs [7,20].

### 4.2. Enhanced Reproductive Fitness Under Cold Stress

Life table analyses further revealed that high light intensity at 12 °C markedly enhanced reproductive performance through multiple pathways: shortening the pre-reproductive period by 46%, extending reproductive duration by 62%, and increasing net reproductive rate (R_0_) and intrinsic rate of increase (r_m_). These light-associated improvements in fecundity and developmental timing suggest that enhanced energy availability under high light mitigates cold-induced metabolic constraints, enabling sustained reproduction despite suboptimal temperatures. The improved early survival under high light further indicates that light-enhanced ATP production supports essential physiological functions under cold stress. This observation illustrates how light availability can directly shape demographic trajectories and life history strategies under thermal challenges.

### 4.3. Ecological and Evolutionary Implications

The temperature-specific effects of light on *A. pisum* fitness highlight a sophisticated adaptation to seasonal environments. The ability to harness solar energy for ATP synthesis may provide a selective advantage in early spring or late autumn, when temperatures are low but sunlight is abundant. This plasticity could explain the predominance of green morphs in cold climates and their rapid population growth under favorable light conditions [7]. Our results are closely consistent with the known photo-physical properties of carotenoids, especially their ability to associate with light-driven electron transfer, which further supports the proposed mechanism [5,7].

## 5. Conclusions

Our results underscore the importance of light as a critical environmental factor that interacts with temperature to shape aphid life history strategies. The light-dependent enhancement of energy metabolism and reproductive output under cold stress provides new insights into the physiological mechanisms underlying insect resilience to climate variability. Understanding these light-associated adaptations could improve predictive models of aphid population dynamics and inform the development of targeted management strategies in temperate agroecosystems.

## Figures and Tables

**Figure 1 insects-16-01013-f001:**
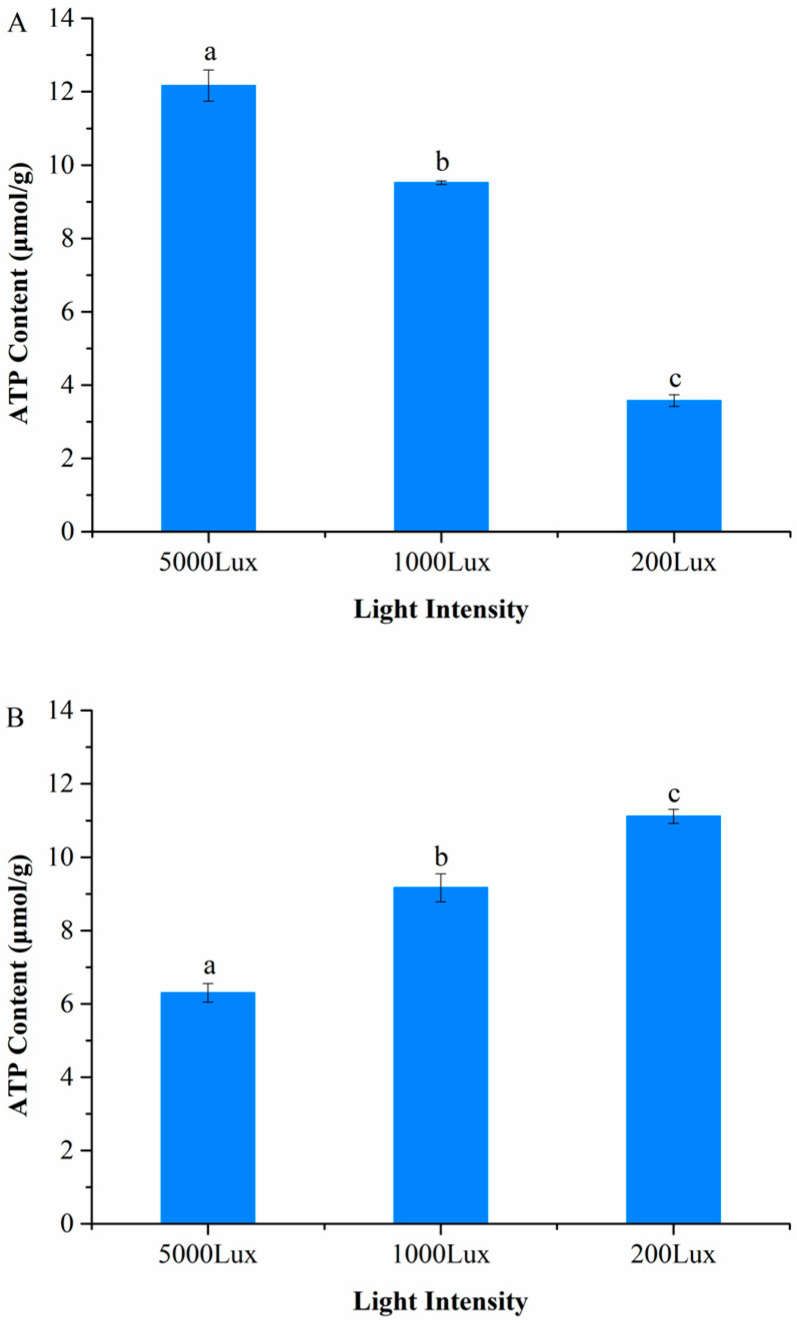
Temperature-dependent effects of light intensity on ATP content in *A. pisum*. (**A**) At 12 °C. (**B**) At 22 °C. Data represent mean ± SEM (*n* = 3 independent biological replicates, each with 10 pooled aphids). Different lowercase letters indicate significant differences among light intensities within the same temperature (Tukey’s HSD test, *p* < 0.05).

**Figure 2 insects-16-01013-f002:**
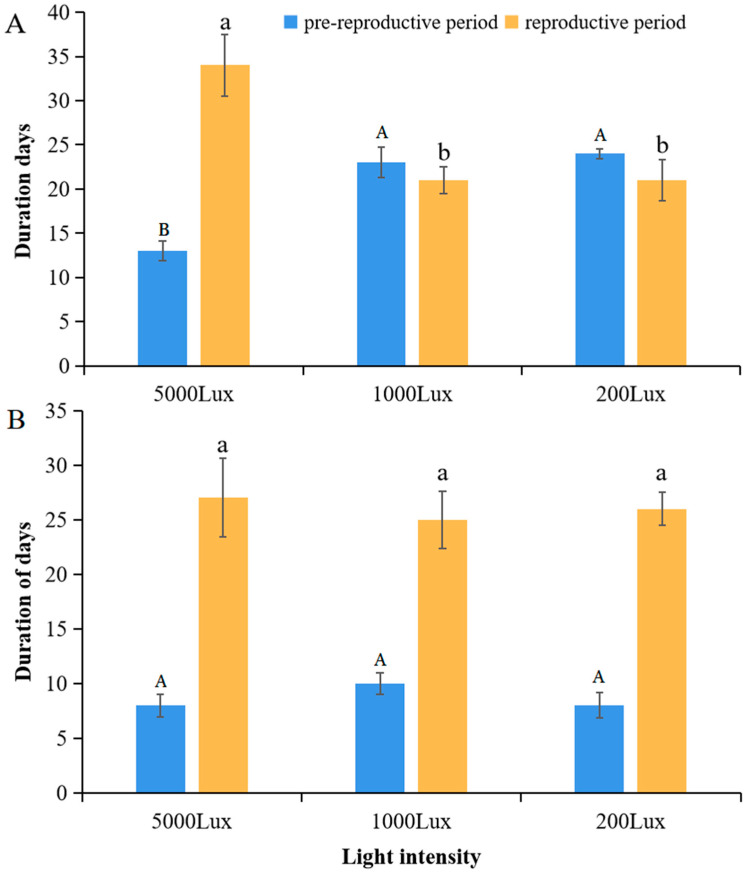
Developmental timing responses to light intensity at (**A**) 12 °C and (**B**) 22 °C. Data represent mean ± SEM (*n* = 3 independent experimental replicates; each replicate included 30 pooled aphids from a single plant). Different letters indicate significant differences among light intensities within the same temperature (Tukey’s HSD test, *p* < 0.05).

**Figure 3 insects-16-01013-f003:**
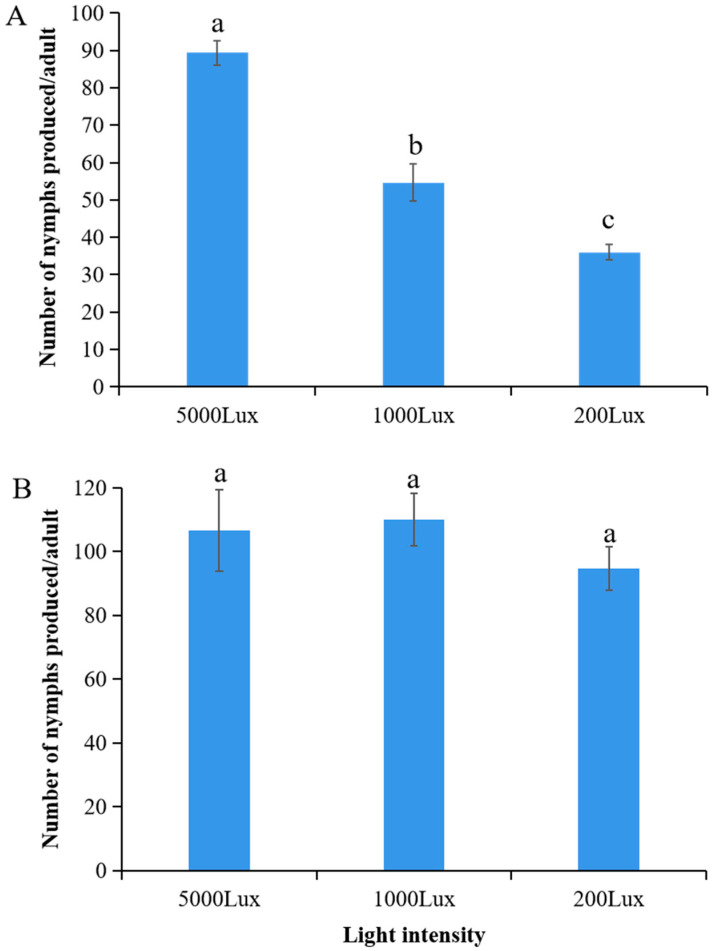
Fecundity of *A. pisum* under varying light intensities at (**A**) 12 °C and (**B**) 22 °C. Data represent mean ± SEM (*n* = 3 independent experimental replicates; each replicate included 30 pooled aphids from a single plant). Different lowercase letters indicate significant differences among light intensities within the same temperature (Tukey’s HSD test, *p* < 0.05).

**Figure 4 insects-16-01013-f004:**
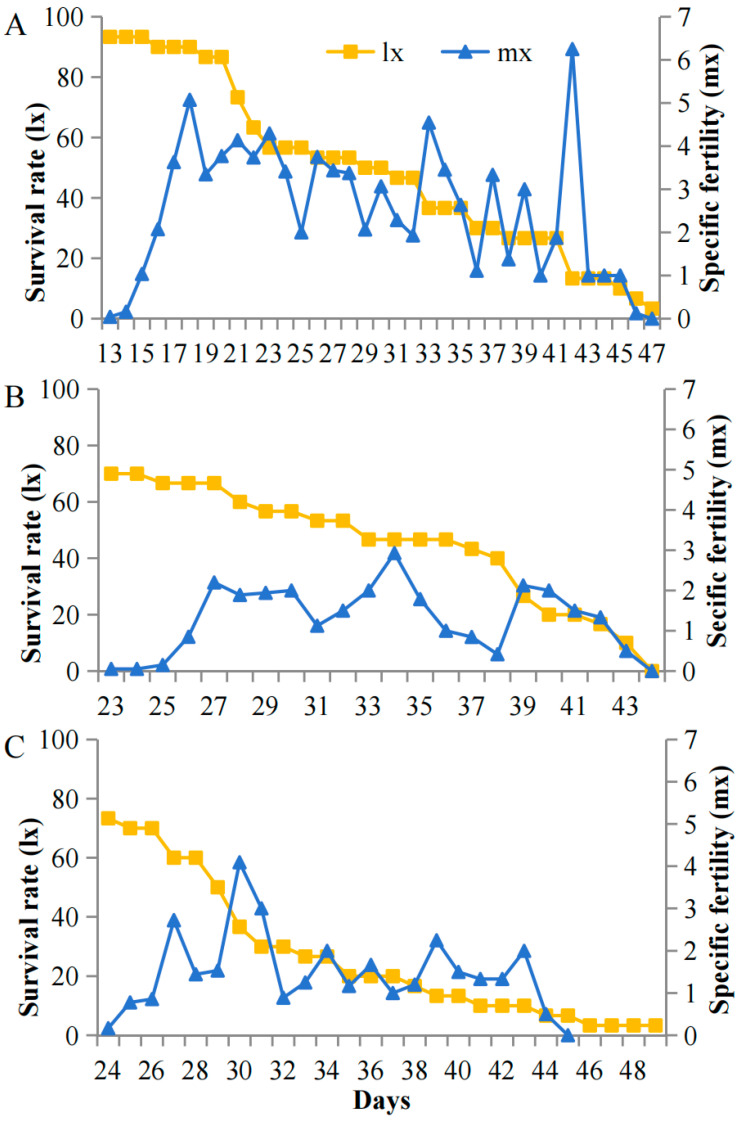
Survival (l_x_) and fecundity (m_x_) schedules of *A. pisum* at 12 °C under (**A**) 5000 lux, (**B**) 1000 lux, and (**C**) 200 lux.

**Figure 5 insects-16-01013-f005:**
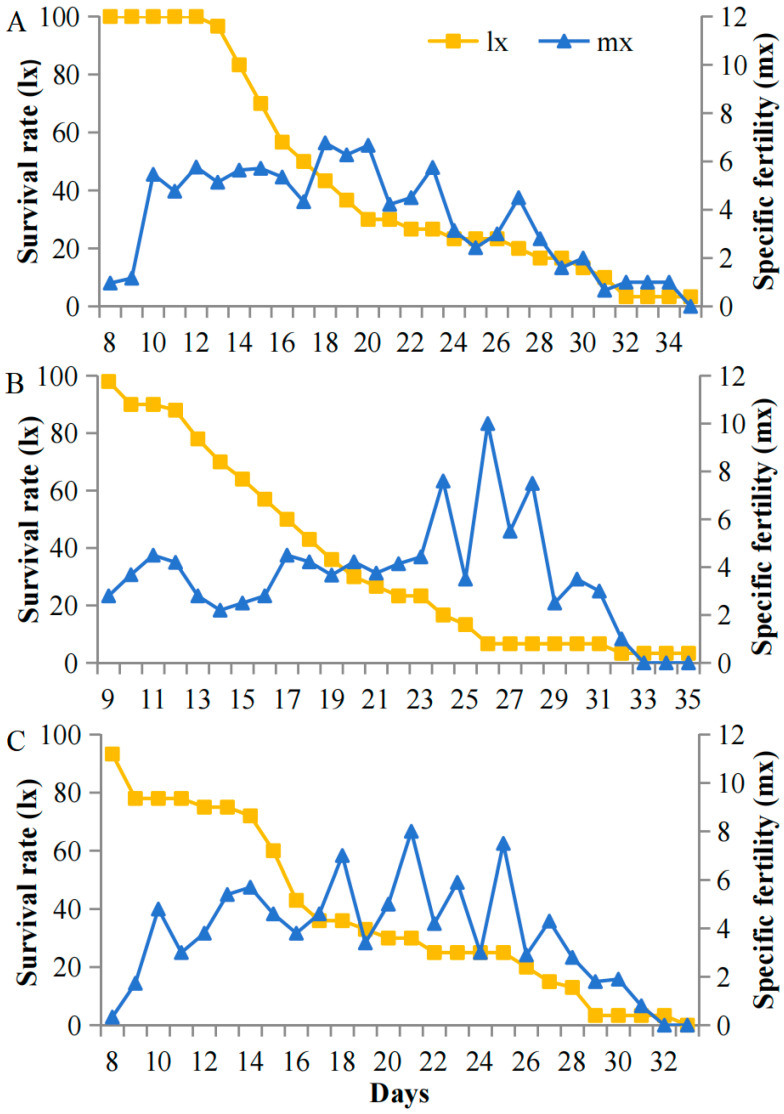
Survival (l_x_) and fecundity (m_x_) schedules at 22 °C under (**A**) 5000 lux, (**B**) 1000 lux, and (**C**) 200 lux.

**Table 1 insects-16-01013-t001:** Life table parameters of *A. pisum* under different light intensities at 12 °C.

Temp.	Light Intensity	Net Reproductive Rate (R_0_) [95% CI]	Intrinsic Rate of Increase (r_m_, day^−1^) [95% CI]	Generation Time (T, Days) [95% CI]	Doubling Time (DT, Days) [95% CI]
12 °C	200 lux	44.1 ± 3.2 [37.2, 51.0] a	0.14 ± 0.01 [0.11, 0.17] a	26.1 ± 1.2 [23.2, 29.0] a	5.5 ± 0.3 [4.8, 6.2] a
	1000 lux	63.7 ± 4.5 [53.8, 73.6] b	0.18 ± 0.01 [0.15, 0.21] b	21.3 ± 0.9 [19.2, 23.4] b	4.1 ± 0.2 [3.6, 4.6] b
	5000 lux	82.3 ± 5.1 [70.2, 94.1] c	0.21 ± 0.01 [0.18, 0.24] c	18.2 ± 0.8 [16.3, 20.1] c	3.3 ± 0.2 [2.9, 3.8] c
	Statistics	F_2,6_ = 19.23, *p* < 0.001	F_2,6_ = 13.74, *p* = 0.002	F_2,6_ = 7.16, *p* = 0.015	F_2,6_ = 27.33, *p* < 0.001
22 °C	200 lux	71.2 ± 5.1 [59.8, 82.6]	0.19 ± 0.01 [0.16, 0.22] a	16.1 ± 0.7 [14.5, 17.7] a	3.7 ± 0.4 [2.8, 4.6] a
	1000 lux	68.5 ± 4.2 [58.9, 78.1] a	0.18 ± 0.01 [0.15, 0.21] a	15.8 ± 0.6 [14.4, 17.2] a	3.9 ± 0.3 [3.2, 4.6] a
	5000 lux	72.3 ± 6.0 [58.1, 86.5] a	0.20 ± 0.01 [0.17, 0.23] a	15.3 ± 0.5 [14.1, 16.5] a	3.5 ± 0.2 [3.0, 4.0] a
	Statistics	F_2,6_ = 0.42, p = 0.67	F_2,6_ = 0.87, *p* = 0.52	F_2,6_ = 0.31, *p* = 0.74	F_2,6_ = 0.25, *p* = 0.78

*Note: Values are presented as mean ± SEM (n = 3 independent experimental replicates; each replicate included 30 pooled aphids from a single plant). Different lowercase letters within the same temperature column indicate significant differences among light intensities (Tukey’s HSD test, *p* < 0.05).*

## Data Availability

The original contributions presented in this study are included in the article. Further inquiries can be directed to the corresponding author.

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
