# Peer review of "Horizontally Transferred Carotenoid Genes Associated with Light-Driven ATP Synthesis to Promote Cold Adaptation in Pea Aphid, Acyrthosiphon pisum"

_insects, 2025, doi:10.3390/insects16101013_

Round 1
Reviewer 1 Report
Comments and Suggestions for Authors
The authors present experiments on the pea aphid Acyrthosiphon pisum that report a temperature-gated, light-dependent increase in ATP content and dramatic life-history benefits at 12°C under high light (5000 lux), and no such effects at 22°C; the work combines short-term biochemical assays (ATP quantification) with cohort life-table experiments and demographic modeling to argue that horizontally acquired carotenoid biosynthesis enables light-harvested ATP to supplement cold-stressed metabolism. The study addresses a novel and important question with potentially high impact for ecological and evolutionary physiology, but crucial methodological and inferential weaknesses substantially weaken the evidence for the central mechanistic claim (carotenoid-mediated photo-ATP synthesis). Overall quality: 50/100. Language quality: 7/10.
The Methods lack detail and contain potentially fatal procedural choices (insufficient replication for biochemical assays, unclear experimental units and potential pseudoreplication, absence of information on light spectrum and microclimate, and use of statistical tests that do not match the factorial design), so confidence in the reported effects is limited; the Discussion overstates causality by attributing effects to carotenoid-mediated electron transfer without any direct manipulation or biochemical evidence (no carotenoid quantification, no genetic or pharmacological perturbation, no measurement of redox intermediates or photochemical activity), and it fails to acknowledge several plausible alternative explanations (microwarming under high light, maternal/host-plant mediated effects, or diurnal timing artifacts).
Major points:
Reported experimental design does not match the statistical approach: the study tests temperature × light effects but uses separate one-way ANOVAs per temperature rather than a two-way ANOVA (or mixed model) that explicitly tests the interaction; without an interaction test the claim of a temperature-gated effect is not statistically demonstrated.
ATP assay methodology is insufficiently described and technically questionable: a commercial ATP kit is reported with fluorescence read at 700 nm and a 37°C/20 min incubation step — luciferase-based ATP assays are typically luminescence based and immediate; the protocol as written risks ATP degradation or artifactual signal and lacks details on standards, detection limits, sample homogenization, and whether readings were normalized to protein vs fresh mass; therefore ATP values and their precision are uncertain.
Low replication and potential pseudoreplication weaken biochemical inference: ATP measures use n=3 biological replicates of ten aphids each (as described), which provides very limited power and may confound within-plant effects; for life-table cohorts the unit of replication is unclear (individual nymphs vs cohort/plant), and the use of three cohorts per treatment likely underestimates variance if multiple individuals were nested on the same plant without accounting for plant as the experimental unit.
Missing control and direct tests of the proposed mechanism: the central causal statement that horizontally transferred carotenoid genes enable photo-ATP synthesis is correlative here — no comparison of carotenoid-deficient (red) vs carotenoid-rich (green) morphs, no RNAi/CRISPR or chemical inhibition of carotenoid biosynthesis, and no direct measurement of carotenoid content, absorbance spectra, photochemical electron transfer, ROS/redox status, or localization of pigments to membranes; without these, the mechanistic interpretation is speculative.
Light environment is inadequately characterized: lux values alone are insufficient because spectral composition (wavelengths available to carotenoids), photon flux, and the light source (LED/incandescent/fluorescent) dramatically affect photochemistry; moreover, there is no reporting of microclimate at the aphid/plant surface — high light treatments may have produced local heating of plants or microhabitats, confounding "light" with temperature.
Life-table parameter estimation and reporting are flawed or incomplete: the method used to estimate rₘ is approximated (rm ≈ ln(R₀)/T) rather than solved from the Lotka–Euler equation or by bootstrap methods that give robust confidence intervals; degrees of freedom reported for ANOVAs (e.g., F₂,₈) are inconsistent with stated replicate numbers, suggesting reporting errors or calculation mistakes that must be corrected.
Overinterpretation of types of survival curves and ecological meaning: terms "Type I" vs "Type II" survival patterns without hazard/function tests or clear sample-size justification are judgmental; ecological meaning of 5000 lux must be placed in context (5000 lux is well short of full sun and spectral quality varies), so extrapolation to field seasonality dynamics requires caution and additional data.
Reproducibility and transparency issues: certain key experimental information are missing (which light source and spectrum, observation of plant microclimate, time of day of sampling, aphid feeding past, how many independent parent plants within the cohorts, how outliers were handled), and raw data or code for demographic analysis are not provided; such omissions make independent testing challenging.
Minor points:
Units and reporting inconsistencies: ATP expressed as μmol ATP/g fresh weight is plausible but high; methods do not describe sample pooling strategy, mass measurement procedures, or limits of detection — correct and clarify units and sample handling.
Statistical reporting errors: several F values and degrees of freedom appear inconsistent with sample sizes (please provide ANOVA tables, residual diagnostics, and exact n per group used to compute each statistic).
Life-table cohort initiation may introduce selection bias: initiating cohorts by placing adults until >30 offspring are produced then removing adults can bias cohort composition (maternal age and condition effects); describe and justify this approach or use truly randomized neonates from multiple mothers.
A few references and DOIs appear duplicated or misformatted.
Figures and legends need improvement: error bars are labelled as SEM but sample sizes are small; include exact n in all figure legends, clarify whether bars are SEM or SD, and provide individual data points where possible.
Minor method clarifications: indicate whether aphids were fasted before ATP assays, whether CO₂/anesthesia was used during handling, and whether assays were performed blind to treatment to minimize bias.
The reported results, if replicated with corrected methodology and direct mechanistic tests, would be exciting and could meaningfully change understanding of non-photosynthetic light utilization and cold-adaptation in arthropods; however, as presented the large effect sizes (2–2.5× changes in ATP and fecundity) rest on methods and statistics that require correction and stronger mechanistic linkage before broad applicability can be endorsed.
Recommendation to Editor: major revision. Require the authors to (a) provide full raw data and ANOVA tables, (b) reanalyze with appropriate factorial statistics and report interaction tests, (c) substantially expand methodological detail (ATP assay validation, light spectra, microclimate measurements, units and pooling strategy), (d) increase replication or demonstrate through power analysis that current replication is adequate, and critically (e) supply direct evidence linking carotenoids to the photo-ATP effect (quantify carotenoids, compare morphs or perform targeted knockdown/inhibition, or present photochemical/redox assays). If authors cannot address these core issues experimentally or with additional analyses, the conclusions should be substantially tempered.
Reviewer 2 Report
Comments and Suggestions for Authors
Dear authors.
The study is interesting, however, are needs several adjust in the document (see details inside document attached)

Reviewer 3 Report
Comments and Suggestions for Authors
A simple, but ecologically important study shows how the abiotic factors light and temperature interact to shape fitness of an important pest insect. The manuscript is professionally written but needs some editorial and spelling corrections:
- line 21: insert a space between activation and of
- line 27: why life-table in uppercase letters?
- line 52 and others: why light-dependent in uppercase letters?
- headline 2.1 and others: all headlines either in uppercase or in lowercase letters, not mixed
- line 81 and others: all species names in italics, also in the references
- line 101: do bars in figures show SE or SEM (see Fig. 1)? Insert a full stop at the end of the sentence
- legend to Fig. 3: say that daily fecundity is shown
- line 165: no shaded regions are shown in the figures
- line 171: a link to Table 2 is missed
- lines 208 to 210: use lowercase letters after commas
- line 217: different letters are not shown within columns, but within rows; significant differences are not shown compared to "control" but to the parameter 200 lux
- References: give all journal titles either in italics or in normal fonts, but not mixed, and all species names in italics
- line 266: insert "USA"
Round 2
Reviewer 1 Report
Comments and Suggestions for Authors
The authors have made a clearly conscientious and substantial effort to address the major statistical and reporting deficiencies noted in the original review. In particular they re-analysed the data with two-way ANOVAs, solved the Euler–Lotka equation numerically for rₘ rather than using the earlier approximation, expanded and clarified many methodological details (ATP protocol, sampling time, light source), removed subjective survival-type classifications, and consolidated life-table reporting to improve clarity. Those are important and appropriate corrections that materially strengthen the manuscript’s internal consistency and transparency.
Partially addressed or unresolved issues (these need further action before acceptance)
Mechanistic evidence remains correlative. The central mechanistic concern—the claim that horizontally transferred carotenoid genes cause light-driven ATP synthesis—was not directly tested. The authors explicitly state they were unable to perform genetic (RNAi/CRISPR) or biochemical experiments (carotenoid quantitation, absorbance/spectral measurements, membrane localization, direct electron transfer assays, ROS/redox measures) and instead rely on citation of earlier work (Valmalette et al. 2012) and a strengthened literature discussion. This is an honest and reasonable narrative approach, but it leaves the mechanistic claim essentially inferential rather than demonstrated. The manuscript must therefore continue to frame the mechanism as hypothesis / plausible interpretation and avoid causal language implying direct proof. If the editors demand mechanistic proof, additional experiments will be required.
Inconsistencies about experimental units and pooling / potential pseudoreplication. The cover letter and methods contain seemingly conflicting descriptions about pooling for ATP assays versus cohort/plant replication. At times the text states each ATP biological replicate is “derived from a separate cohort and plant,” while elsewhere the authors say ATP replicates were pooled from multiple plants. This ambiguity leaves unresolved whether the true experimental unit for biochemical assays was the plant, the cohort, or an improperly pseudo-replicated pool of individuals. The authors must (a) state unambiguously the biological experimental unit for each analysis, (b) confirm whether samples were pooled across plants within a replicate or not, and (c) re-compute statistics if necessary (and transparently report how pooling affects degrees of freedom). Without that clarification, the risk of pseudoreplication remains.
Residual reporting inconsistencies in ANOVA degrees of freedom and statistics. Although the authors say they corrected degrees of freedom and re-ran two-way ANOVAs, the manuscript still displays inconsistent F and df values between text and table(s) (e.g., F(2,12) reported for some analyses in the Results but F₂,₆ or other dfs shown in Table 1). These internal inconsistencies must be resolved: every reported statistic should match the underlying ANOVA tables. I requested ANOVA tables and residual diagnostics; the authors state re-analysis was performed but have not (as far as I can see) provided full ANOVA tables, residual plots/diagnostics, or model assumptions checks in a supplement. The authors should attach the ANOVA tables, model residual diagnostics, and exact n used for each test as supplemental material (or include them in the manuscript) so readers can verify the analyses.
Sample size, power and uncertainty estimates for demographic parameters. The authors defend their small n (n = 3 replicates) by producing a post-hoc power statement (power = 0.92), but this does not eliminate concerns about biological variability, low replication and the robustness of SEM with small n. For demographic parameters like rₘ, it is standard to provide bootstrap confidence intervals (or other robust CIs) because the distribution of life-table estimates is non-normal and sample sizes here are small. The manuscript reports point estimates ± SEM but I did not find bootstrap CIs for rₘ, R₀, or DT. The authors should (and in many journals must) present bootstrap (or parametric) CIs for rₘ and R₀ and, ideally, report effect sizes with CIs rather than only SEMs. This will substantially increase confidence in the demographic inferences.
Light characterization and microclimate. Specifying “cool white LED, 400–780 nm” is better than nothing, but lux alone is an inadequate measure for photochemistry. Carotenoid photophysics depend on spectral irradiance and photon flux density (μmol photons m⁻² s⁻¹) in relevant wavelengths; lux hides spectral differences and is weighted to human vision. The authors should report the measured spectral irradiance / spectral power distribution of the lamps (a plot or tabulated spectra), and the photon flux in the carotenoid-responsive band. They should also present microclimate data at the plant/aphid surface (leaf surface temperature, any local heating under the lamps) to rule out confounding light with micro-heating. At minimum these measurements should be provided in a supplement; without them, interpretation that effects are photochemical rather than microclimate driven remains only partially supported.
Other smaller issues to fix before acceptance
Explicitly temper causal language in the abstract and conclusions (change “enable” / “mediate” to “may enable” / “are consistent with” unless the authors add mechanistic tests).
Clarify the exact ATP units and justify whether μmol ATP g⁻¹ FW values are biologically plausible (a short note comparing to published ATP concentrations in small insects would help).
Reviewer 2 Report
Comments and Suggestions for Authors
dear authors, the manuscript has improved considerably, but please consider addressing some additional issues (as previously suggested). see attached document
